# Hidden Pool of Cardiac Adenine Nucleotides That Controls Adenosine Production

**DOI:** 10.3390/ph16040599

**Published:** 2023-04-15

**Authors:** Magdalena A. Zabielska-Kaczorowska, Alicja Braczko, Iwona Pelikant-Malecka, Ewa M. Slominska, Ryszard T. Smolenski

**Affiliations:** 1Department of Physiology, Medical University of Gdansk, 80-210 Gdansk, Poland; 2Department of Biochemistry, Medical University of Gdansk, 80-210 Gdansk, Polandryszard.smolenski@gumed.edu.pl (R.T.S.); 3Division of Medical Laboratory Diagnostics, Medical University of Gdansk, 80-210 Gdansk, Poland; 4Heart Science Centre, Imperial College at Harefield Hospital, Harefield UB9 6JH, UK

**Keywords:** adenosine, rat heart, ischemia, contractility, ATP, nucleotide catabolites, 31P NMR, mitochondria

## Abstract

Myocardial ischemic adenosine production decreases in subsequent events that may blunt its protective functions. To test the relation between total or mitochondrial cardiac adenine nucleotide pool (TAN) on the energy status with adenosine production, Langendorff perfused rat hearts were subjected to three protocols: 1 min ischemia at 40 min, 10 min ischemia at 50 min, and 1 min ischemia at 85 min in Group I; additional infusion of adenosine (30 µM) for 15 min after 10 min ischemia in Group I-Ado, and 1 min ischemia at 40 and 85 min in the controls (Group No I). A ^31^P NMR and an HPLC were used for the analysis of nucleotide and catabolite concentrations in the heart and coronary effluent. Cardiac adenosine production in Group I measured after 1 min ischemia at 85 min decreased to less than 15% of that at 40 min in Group I, accompanied by a decrease in cardiac ATP and TAN to 65% of the initial results. Adenosine production at 85 min was restored to 45% of that at 40 min in Group I-Ado, accompanied by a rebound of ATP and TAN by 10% vs. Group I. Mitochondrial TAN and free AMP concentrations paralleled that of total cardiac TAN. Changes in energy equilibrium or mitochondrial function were minor. This study highlights that only a fraction of the cardiac adenine nucleotide pool is available for adenosine production, but further studies are necessary to clarify its nature.

## 1. Introduction

Adenosine is a purine catabolite produced in the heart that has long been recognized as a coronary vasodilator [1,2,3,4]. Its other cardioprotective activities include antiadrenergic, antiplatelet, antiproliferative, and antileukocyte actions, as well as an important role in angiogenesis and preconditioning [5,6,7,8]. It is therefore of interest in heart surgery and cardiology, since the regulation of adenosine concentration in the heart could be a therapeutic strategy to reverse the effects of myocardial ischemia [9,10,11]. An increase in postischemic myocardial performance was demonstrated experimentally when exogenous adenosine was applied either before or after ischemia or when endogenous adenosine production was increased [12,13].

Adenosine production and nucleotide catabolism are known to be directly related to the energy status of the cell; their rates increase when the cell becomes energy depleted [14,15,16,17,18]. However, it is not clear which other factors contribute to the regulation of nucleotide catabolism and adenosine production. A significant role of 5’-nucleotidase and adenosine monophosphate deaminase (AMPD) regulation has been raised in several reports, including the role of adenosine prephosphorylation by adenosine kinase, which has been implicated in the overall balance of adenosine release [19,20,21,22,23,24,25,26,27].

In our previous studies, we have shown that adenosine and other purine catabolites are released from the heart during reperfusion after prolonged ischemia, and after the purine catabolite washout phase, they decrease to below preischemic levels. In addition, we determined that subsequent brief ischemic events result in reduced purine release [28]. Moreover, we found that the mechanism of this phenomenon is linked to changes in energy metabolism (phosphocreatine overshoot and increased phosphorylation potential in the postischemic heart) [29]. We also discovered that a transient supply of exogenous adenosine during reperfusion could boost endogenous adenosine production in the postischemic heart, which was linked to adenine nucleotide pool restoration [30]. However, analysis of the effects of mitochondrial compartmentation of adenine nucleotides and their function on adenosine production and nucleotide catabolism has not been performed. In this study, we analyzed the adenine nucleotide pool in the heart and mitochondria, as well as the estimated free AMP concentration, to evaluate the relation between purine catabolite release and adenine nucleotide concentration in the heart before and after global ischemia with and without a supply of adenosine during reperfusion.

## 2. Results

### 2.1. Release of Purine Catabolites and Adenosine in Coronary Effluent

Figure 1a presents the concentrations of purine catabolites (the sum of adenosine, inosine, hypoxanthine, xanthine, and uric acid concentrations) and adenosine before and after the 1 min ischemic events at 40 and 85 min of the experiment. The purine catabolite release was maintained in Group No I. However, in Group I, virtually no purine catabolite increase was observed after 1 min ischemia at 85 min of the experiment. The increase in purine was restored in Group I-Ado, and catabolite concentrations accounted for approximately 30% of the increase observed after 1 min ischemia at 40 min of the experiment. Adenosine concentration changes followed a similar pattern (Figure 1b).

### 2.2. ATP (Adenosine Triphosphate), ADP (Adenosine Diphosphate), AMP (Adenosine Monophosphate), NAD (Nicotinamide Adenine Dinucleotide), Oxidized and Reduced Form of NAD (NADH), Total Adenine Nucleotide Pool (TAN), and Total NAD Pool (TNAD) in the Heart

Figure 2a shows the concentrations of adenine nucleotides in the heart at the end of the experiment. These data show a decrease in ATP concentration and depletion of the adenine nucleotide pool in Group I and its partial restoration in Group I-Ado. There was no significant difference between the hearts in terms of the myocardial content of other energy metabolites (Figure 2a,b).

### 2.3. Mitochondrial ATP, ADP, AMP, NAD, NADH, TAN, and TNAD in the Heart

Figure 3a presents the concentrations of adenine nucleotides in the mitochondria of the rat heart at the end of the experiment. These results indicate that the mitochondrial pool remained in equilibrium with total cardiac nucleotide concentration. Depletion of the cardiac adenine nucleotide pool results in depletion of the mitochondrial pool, and its restoration results in recovery of the mitochondrial pool as well (Figure 2a and Figure 3a). A significant decrease in ATP, ADP, and total adenine nucleotide content in Group I was found (Figure 3a). Figure 3b shows a decrease in NAD and TNAD in the mitochondria isolated from Group I and a similar concentration in Group I-Ado when compared to C hearts.

### 2.4. Concentration of Free AMP Calculated Based on ^31^P NMR (Nuclear Magnetic Resonance) Spectra

The ^31^P NMR spectra allowed the estimation of ATP, phosphocreatine inorganic phosphate concentrations, and pH value throughout the experiment. ATP levels remained constant during the 1 min ischemia at 40 and 85 min of the experiment, while they were lower in hearts subjected to 10 min ischemia in Group I. A transient infusion of adenosine during reperfusion after 10 min of ischemia partially restored ATP levels in Group I-Ado (Figure 4a). Phosphocreatine concentration decreased by 50% during 1 min ischemia at 40 min and was similar in all Groups. It was increased, however, before ischemia at 85 min in Groups I and I-Ado, but not in Group I. During 1 min ischemia at 85 min, phosphocreatine concentrations in Groups I and I-Ado were slightly higher than in No I (Figure 4b). Intracellular inorganic phosphate increased two-fold during ischemia at 40 min in Group I, followed by a rapid decrease to below preischemic values at reperfusion. These changes were similar in all Groups. This pattern of changes in inorganic phosphate concentration during 1 min ischemia were similar in all Groups at both time points (Figure 4c).

The concentrations of ATP, phosphocreatine, and inorganic phosphate, and the pH value obtained from the ^31^P NMR spectra, were used to calculate the free AMP concentration. During 1 min ischemia at 40 min, the free AMP concentration (Figure 5a) increased to 8–11 μM. This increase was reduced to the same extent (4 and 5 μM, respectively) in Groups I and I-Ado during 1 min ischemia at 85 min, while it was maintained at 9 μM in Group No I.

The pH decreased from a preischemic value of 7.04 to 6.90 during ischemia after 40 min of perfusion. This change was similar in all Groups. However, during 1 min ischemia at 85 min of perfusion, pH decreased only to 6.96 in the I and I-Ado Groups, while the magnitude of the pH decrease was maintained in Group No I (Figure 5b).

### 2.5. Oxygen Consumption in the Mitochondria of Rat Hearts

Figure 6 shows the oxygen consumption rate in the mitochondria of the rat heart subjected to repeated 1 min and 10 min ischemic intervals, as specified above. The oxygen consumption rate in complex I of the isolated mitochondria was increased in the I-Ado Group in comparison with the initial concentration in the No I and I Groups.

The mechanical function recorded using the intraventricular balloon was similar in all Groups at the start of the experiments. The left ventricular developed pressure was maintained above 95% of the initial values in Group No I, and complete recovery was observed after 10 min ischemia in the Groups I and I-Ado before 1 min ischemia at 85 min of the experiment. No differences in heart rate or coronary flow were observed between the Groups or between 40 and 85 min of the experiment.

## 3. Discussion

This study demonstrates that the decrease in adenosine and other nucleotide catabolite production in the hearts subjected to repeated ischemic events could be related to depletion of the adenine nucleotide pool and not to changes in the regulation of energy metabolism. We observed a considerable increase in endogenous purine catabolite production after the transient infusion of exogenous adenosine, which was associated with an increase in the cardiac adenine nucleotide pool. However, the profound decrease in adenosine production, with only a small depletion of the cardiac nucleotide pool, suggests that the entire pool is not available for nucleotide catabolism. Our further analyses argue against the sequestration of adenine nucleotides in mitochondria.

Because adenosine, one of the purine catabolites, is involved in numerous cardioprotective mechanisms, the regulation of its production has been the focus of numerous studies [1,21,31,32]. Nucleotide breakdown rate, and particularly cytosolic 5’-nucleotidase activity, have been primarily related to cytosolic free AMP concentration, reflecting the energy status of the cell. In this way, when cells become energy-deficient due to insufficient oxygen supply, AMP concentration increases, leading to enhanced adenosine and other catabolite production. However, other factors, such as regulation of cytosolic 5’-nucleotidase and AMP deaminase by inorganic phosphate concentration and pH, have been raised in other studies [33,34,35]. Our results confirmed that purine release after 1 min is markedly reduced if preceded by a longer ischemic period and reperfusion [36,37,38]. Using NMR spectroscopy, we identified several metabolic changes that could be responsible for this change in nucleotide catabolism and the increased adenine nucleotide levels caused by transient adenosine infusion.

Phosphocreatine overshoot and the consequent increase in the phosphorylation potential are closely related to the free cytosolic AMP concentration and, hence, purine catabolite production [39,40]. Changes in the regulation of energy metabolism and stimulation of high energy phosphate generation are the likeliest explanations for the mechanism of phosphocreatine overshoot observed in our study before 1 min ischemia at 85 min in Groups I and I-Ado. Since the left ventricular developed pressure recovered to preischemic levels after 10 min ischemia, there were no differences in heart workload between 40 and 85 min of experiments, so decreased energy utilization due to decreased mechanical work cannot account for the observed difference. This change in energy metabolism was particularly clear in our experiments, since the hearts were perfused at high calcium concentrations with glucose as the only substrate, which resulted in a lower baseline phosphocreatine/ATP ratio and lower phosphorylation potential values [28].

The attenuation of intracellular acidosis during 1 min ischemia at 85 min of the experiment is another important difference we observed, which may affect the rate of purine catabolite production. However, it is difficult to establish exactly what effect it will have on purine catabolite production. Acidosis has been shown to inhibit nucleotide breakdown enzymes [41]. Thus, the attenuation of acidosis may potentially increase the rate of nucleotide breakdown, an opposite effect to that observed in the present study. However, since this trend was established in severely acidotic hearts, it is possible that the mild acidosis that we observed produces the opposite effect. Details of the effect of varying the degree of acidosis upon purine catabolite production still need to be established, but our results suggest that moderate acidosis causes either inhibition or no effect on nucleotide degradation. Since change of pH during 1 min ischemia at 85 min was similar for Groups with (I-Ado) and without adenosine infusion (I), while we observed a 3-fold increase in adenosine production in Group I-Ado, this pH change is unlikely to be a key factor controlling adenosine production.

The importance of nucleotide pool depletion in decreasing nucleotide catabolite production was previously demonstrated to be significant [42,43]. However, only the present report clearly shows that the increase in nucleotide catabolite production after 1 min ischemia, following a transient supply of adenosine, was not due to changes in other high-energy phosphate metabolites, and that an increase in the ATP and adenine nucleotide pools was the only change observed. An important question is whether this increase causes only a proportional increase in all adenine nucleotide concentrations, including free cytosolic AMP, and whether this results in an accelerated breakdown rate. However, the results of the present study, indicating that a small decrease in nucleotide pool causes a manifold increase in nucleotide breakdown rate in the absence of any other changes, suggest that the relationship is more complex; it may involve changes in enzyme regulation or substrate compartmentation [33,35].

Mitochondrial compartmentation is the most obvious location for the hidden nucleotide pool [44,45]. However, our analysis of the mitochondrial pool argues against mitochondrial involvement in nucleotide pool sequestration in the postischemic heart because it appears to relate to the total nucleotide pool. Furthermore, changes in mitochondrial function were an unlikely factor involved, as none of the changes correlated with adenosine production capacity. Increased activity of complex I in the respiratory chain was noted in the mitochondria when hearts were treated with 10 min ischemia plus an infusion of 30 μM adenosine (I-Ado) during reperfusion. This may relate to the activation of the A3 receptor, which was previously found to affect mitochondrial function [46].

A reduced capacity to produce purine catabolites, particularly adenosine, in the heart after ischemia could have important physiological implications. Since adenosine appears to be a metabolic signal that “retaliates” against external stimulation, preventing myocardial cell damage, a reduction in the ability of myocardial cells to produce adenosine may be associated with impaired endogenous cellular protection [10,43,47]. Stimulation by catecholamines or increased workload may have deleterious effects. The antiplatelet and anti-leukocyte defense mechanisms and regulation of cell proliferation could also deteriorate.

## 4. Materials and Methods

### 4.1. Heart Perfusion

Male Wistar rats weighing 250–300 g were used for our experiments. The maintenance of animals and the experiments conformed with the European Convention for the Protection of Vertebrate Animals used for Experimental and other Scientific Purposes (Council of Europe No. 123, Strasbourg 1985). The rat hearts were excised under deep anesthesia and placed in ice-cold buffer solution, immediately attached to a Langendorff perfusion system, and perfused with filtered (0.45 μm pore size) Krebs–Henseleit buffer solution at constant pressure equivalent to 100 cm of water at 37 °C, as described previously [48]. The buffer solution contained 118 mM NaCl, 4.7 mM KCl, 1.2 MgSO_4_, 1.2 mM KH_2_PO_4_, 0.5 mM EDTA, 24 mM NaHCO_3_, 11 mM glucose, 2.5 mM CaCl_2_, pH 7.4. The perfusate was continuously gassed with a 95% O_2_/5% CO_2_ mixture. Coronary flow was measured by the timed collection of the coronary effluent.

### 4.2. Experimental Protocol

Hearts were subjected to one of three ischemic protocols (Figure 7). Ischemia was performed with cessation of the perfusion buffer flow. After 40 min perfusion to stabilize them, the hearts in all Groups were subjected to 1 min global (37 °C) ischemia followed by 10 min reperfusion. Then, in Groups I and I-Ado, hearts were subjected to 10 min global ischemia followed by reperfusion. In Group I-Ado, adenosine at a concentration of 30 μM was infused for the first 15 min of reperfusion. Then, 25 min after 10 min ischemia, another 1 min ischemia was applied. In controls not subjected to 10 min ischemia (No I), 1 min ischemia was applied at 40 and 85 min. Mechanical function designation was performed using a balloon catheter inserted into the rat heart left ventricle to determine systolic pressure and end-diastolic pressure–volume relations. The balloon was loaded with water from 0 to 200 μL in 25 μL steps. End-diastolic pressure inside the balloon was adjusted to 5 mmHg at the beginning of the experiment and maintained at that level in Group No I, while in the I and I-Ado Groups, the balloon was deflated during 10 min ischemia to avoid endocardial damage, which could affect purine release measurements. End-diastolic pressure was reset to 5 mmHg immediately after 10 min ischemia and remained at that level until the end of the experiment. A Biopac Systems MP 100 connected to a PC running AcqKnowledge 3.7.2 Biopac software (Biopac Systems, Inc., Goleta, CA, USA) was used for data acquisition and calculation. To obtain a value for the initial nucleotide content and mitochondrial isolation, additional hearts (C) were perfused for 20 min and then used for mitochondria isolation or frozen. In all Groups, coronary effluent was collected for 1 min before and after 1 min ischemia. The volumes of the collected fractions were measured, and portions were frozen in liquid nitrogen for analysis of nucleotide catabolite concentrations. At the end of perfusion, the apical part of the hearts was cut for mitochondria isolation and mitochondrial nucleotide pool analysis, and the remaining part was frozen with aluminum clamps precooled in liquid nitrogen for total heart nucleotide pool analysis. Mitochondria isolation was performed according to the previous protocol [49].

### 4.3. Tissue Extraction for High-Performance Liquid Chromatography (HPLC) Analysis

Tissue extracts were prepared from freeze-dried hearts with 0.6 M perchloric acid (25 µL/mg dry tissue). The extracts were then centrifuged (13,000× *g* for 3 min at 4 °C), and the supernatant was neutralized with 2 M KOH. Coronary effluent samples were analyzed without extraction. The analysis was performed with HPLC using a Merck-Hitachi chromatograph (Germany–Japan). The reversed-phase method for the simultaneous determination of purine nucleotides, nucleosides, and bases applied to effluent and tissue extracts has been described previously [50]. Isolated mitochondria were resuspended in 200 μL of 0.4 M HClO_4_ and frozen at −80 °C for at least 24 h. After thawing on ice, mitochondria extracts were collected in Eppendorf-type tubes and neutralized to pH 5.5–6.0 with 3M K_3_PO_4_ and centrifuged. Supernatants were analyzed by HPLC according to the above procedure. Protein precipitates were dissolved in 1.5 mL 0.5 M NaOH and analyzed with the Bradford method [51].

### 4.4. ^31^P NMR Spectroscopy

A similar protocol was used for the perfusion experiment inside the bore of an NMR spectrometer (Bruker AMX-400 wide bore vertical system, ^31^P frequency 161.9 MHz, Germany) to measure changes in myocardial ATP, phosphocreatine (PCr), inorganic phosphate, and pH as we have previously described [41,52]. Fully relaxed spectra were acquired at 20 min of normoxic perfusion (24 scans, 90° angle, and 20 s interpulse delay). Throughout the experiment, saturated spectra (48 scans, 90° angle, and 0.9 s interpulse delay) were collected. An initial ATP concentration of 22.6 µmol/g dry wt, as measured by HPLC, was used for calibration of the NMR data. For calculation of other phosphate metabolite concentrations in saturated spectra, saturation factors obtained from repeated, fully relaxed, and saturated spectra acquired during baseline conditions were applied. Peak height was used for the calculation of ATP and phosphocreatine concentrations, while peak area was used to quantify inorganic phosphate concentrations. The pH was calculated from the chemical shift of the intracellular inorganic phosphate vs. phosphocreatine peak. Free AMP concentration was calculated as described previously [53].

### 4.5. Mitochondrial Respiration Analysis

The electron flow assay was performed as described previously [54]. Briefly, isolated mitochondria from a rat’s heart were suspended in a mitochondrial assay solution (MAS, 1×), which contained 70 mM sucrose, 220 mM mannitol, 10 mM KH_2_PO_4_, 1 mM ethylene glycol-bis(β-aminoethyl ether)-*N*,*N*,*N*′,*N*′-tetraacetic acid tetrasodium salt (EGTA), 5 mM MgCl_2_, 2 mM 4-(2-Hydroxyethyl)piperazine-1-ethanesulfonic acid sodium salt, *N*-(2-Hydroxyethyl)piperazine-*N*′-(2-ethanesulfonic acid) sodium salt (HEPES), and 0.2 % (*w*/*v*) albumin bovine serum (BSA), pH 7.2. To maintain adequate mitochondrial function, the organelles were suspended in an ice-cold MAS buffer supplemented with 4 μM carbonyl cyanide-4-(trifluoromethoxy)phenylhydrazone (FCCP) (Cat. No C2920, Sigma-Aldrich, Saint Louis, MO, USA), 2 mM malate (Cat. No M9138, Sigma-Aldrich, Saint Louis, MO, USA), and 10 mM sodium pyruvate (Cat. No P5280, Sigma-Aldrich, Saint Louis, MO, USA). Mitochondria were diluted to a concentration of 4 µg mitochondrial protein/25 µL of suspension. Mitochondrial protein was determined using the Bradford method. Twenty-five μL of the mitochondrial suspension were applied to the plate and centrifuged at 2000× *g*, 4 °C for 15 min. After centrifugation, 155 μL of warm MAS buffer supplemented with substrates were added to the wells with mitochondria. Then 180 μL of MAS buffer + substrates were added to the empty wells of the plate. The plate was then incubated for 8 min in a non-CO_2_ incubator at 37 °C. In the meantime, the cartridge was refilled in the following order: Port A: 20 μL of 20 μM rotenone (2 μM final) (Cat. No R8875, Sigma-Aldrich, Saint Louis, MO, USA), Port B: 22 μL of 100 mM succinate (10 mM final) (Cat. No S3674, Sigma-Aldrich, Saint Louis, MO, USA), Port C: 24 μL of 40 μM antimycin A (4 μM final) (Cat. No A8674, Sigma-Aldrich, Saint Louis, MO, USA), and Port D: 26 μL of 100 mM ascorbate (Cat. No A92902, Sigma-Aldrich, Saint Louis, MO, USA) plus 1 mM N1, N1, N1, N1-tetramethyl-1,4-phenylene diamine (TMPD) (Cat. No T7394, Sigma-Aldrich, Saint Louis, MO, USA) (10 mM and 100 μM final). Subsequently, the plate was transferred to the Seahorse Analyzer (Agilent Technologies, Santa Clara, CA, USA) and the experiment was initiated. Respiration of Complex II was measured as the succinate-driven OCR. Complex I as well as Complex IV respiration were calculated as the resulting OCR after subtraction of rotenone or antimycin a driven respiration, respectively.

### 4.6. Statistics

GraphPad Prism 9.0 was used for drawing and statistical analysis. Data were shown as mean SEM in this experiment. Values are presented as mean ± SEM. Statistical parametric comparison was performed using one-way analysis of variance (ANOVA) followed by the Student–Newman–Keuls test. Data within one Group at different time points were compared with a paired Student *t*-test with Bonferroni correction for repeated measures. A value of *p* < 0.05 was considered a significant difference.

## 5. Conclusions

Adenosine output in response to a 1 min ischemic event decreased if preceded by 10 min ischemia. An infusion of adenosine during reperfusion after 10 min ischemia was associated with an increase in adenosine production after 1 min ischemia. This restoration of adenosine production capacity was associated with an increase in the myocardial adenine nucleotide pool. Changes in the mitochondrial adenine nucleotide pool essentially followed changes in the total cardiac pool. Endogenous purine catabolite production after the infusion of adenosine was associated with an increase in the adenine nucleotide pool but not with any other changes in energy metabolism. No changes in mitochondrial respiratory chain function were noticed.

These results show that, although changes in energy metabolism are the key factor that regulates adenosine and other purine production in the heart, even a slight change in the nucleotide pool considerably affects purine production without changes in energy metabolism. This may have important implications for the pharmacological regulation of endogenous adenosine production. Therefore, understanding the mechanisms of nucleotide catabolism and adenosine production and studying the restoration of these mechanisms may help to develop novel cardioprotective strategies.

## Figures and Tables

**Figure 1 pharmaceuticals-16-00599-f001:**
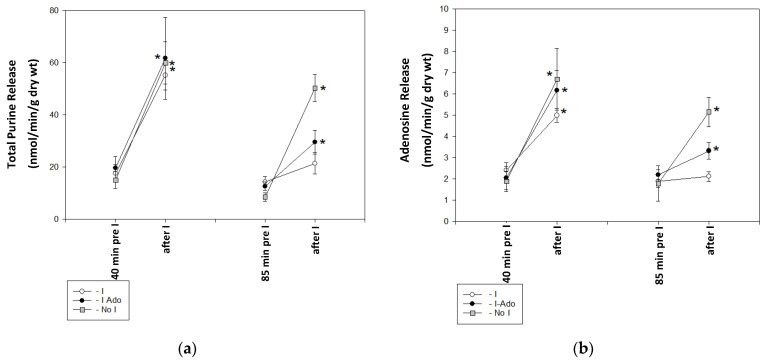
Release of (**a**) purine catabolites (sum of adenosine + inosine + hypoxanthine + xanthine + uric acid) and (**b**) adenosine in coronary effluent before 1 min ischemia and in the effluent collected over 3 min of reperfusion after 1 min ischemia in hearts subjected to 1 min ischemia at 40 min of perfusion, 10 min of ischemia at 50 min of perfusion, and 1 min ischemia at 85 min of perfusion without (Group I) or with (Group I-Ado) a supply of adenosine at 30 µM concentration during the first 15 min of reperfusion and in control hearts (Group No I) subjected three times to 1 min ischemia at 40, 50, and 85 min of perfusion. Values are mean ± SEM (*n* = 5–9). * *p* < 0.05 vs. concentration before 1 min ischemia.

**Figure 2 pharmaceuticals-16-00599-f002:**
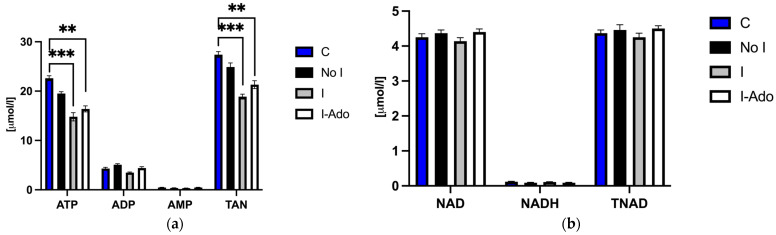
(**a**) ATP, ADP, AMP, and TAN and (**b**) NAD, NADH, and TNAD before, during, and after 1 min ischemia in hearts subjected to 1 min ischemia at 40 min of perfusion, 10 min ischemia at 50 min of perfusion, and 1 min ischemia at 85 min of perfusion without (Group I) or with (Group I-Ado) supply of adenosine at 30 µM concentration during the first 15 min of reperfusion and in control hearts (Group No I) subjected three times to 1 min ischemia at 40, 50, and 85 min of perfusion. For the initial values, control hearts (Group C) were perfused for 20 min. Values are mean ± SEM (*n* = 5–9). ** *p* < 0.01, *** *p* < 0.005.

**Figure 3 pharmaceuticals-16-00599-f003:**
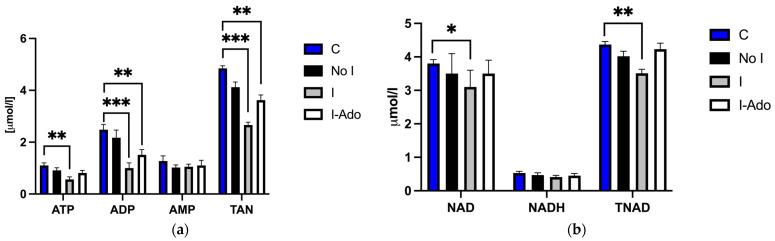
Mitochondrial (**a**) ATP, ADP, AMP, and TAN and (**b**) NAD, NADH, and TNAD before, during, and after 1 min ischemia in hearts subjected to 1 min ischemia at 40 min of perfusion, 10 min ischemia at 50 min of perfusion, and 1 min ischemia at 85 min of perfusion without (Group I) or with (Group I-Ado) supply of adenosine at 30 µM concentration during the first 15 min of reperfusion and in control hearts (Group No I) subjected three times to 1 min ischemia at 40, 50, and 85 min of perfusion. For the initial values, control hearts (Group C) were perfused for 20 min. Values are mean ± SEM (*n* = 5–9). * *p* < 0.05, ** *p* < 0.01, *** *p* < 0.005.

**Figure 4 pharmaceuticals-16-00599-f004:**
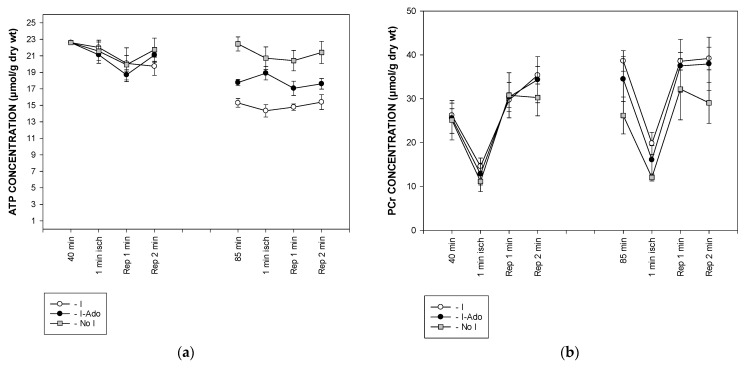
The ^31^P NMR spectra based calculation of (**a**) ATP, (**b**) phosphocreatine, and (**c**) intracellular inorganic phosphate concentrations before, during, and after 1 min ischemia in hearts subjected to 1 min ischemia at 40 min of perfusion, 10 min ischemia at 50 min of perfusion, and 1 min ischemia at 85 min of perfusion without (Group I) or with (Group I-Ado) supply of adenosine at 30 µM concentration during the first 15 min of reperfusion and in control hearts (Group No I) subjected three times to 1 min ischemia at 40, 50, and 85 min of perfusion. For the initial values, control hearts (Group C) were perfused for 20 min. Values are mean ± SEM (*n* = 5–9).

**Figure 5 pharmaceuticals-16-00599-f005:**
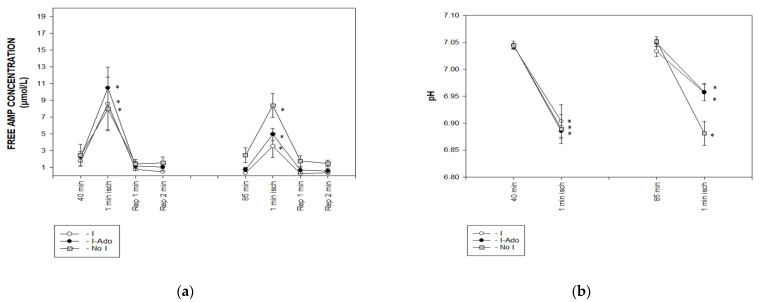
The ^31^P NMR spectra based calculation of (**a**) free AMP concentration and (**b**) pH before, during, and after 1 min ischemia in hearts subjected to 1 min ischemia at 40 min of perfusion, 10 min ischemia at 50 min of perfusion, and 1 min ischemia at 85 min of perfusion without (Group I) or with (Group I-Ado) supply of adenosine at 30 μM concentration during the first 15 min of reperfusion and in control hearts (Group No I) subjected three times to 1 min ischemia at 40, 50, and 85. Values are mean ± SEM (*n* = 5–9). * *p* < 0.05 vs. value before ischemia.

**Figure 6 pharmaceuticals-16-00599-f006:**
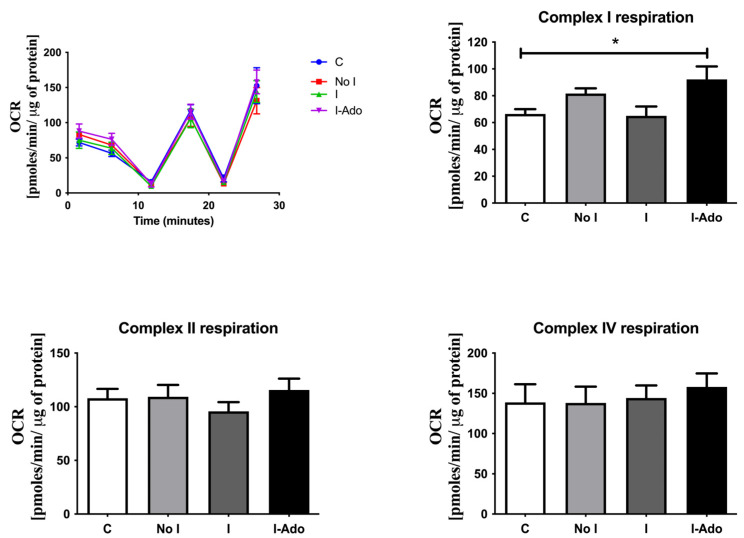
Oxygen consumption rate (OCR) in the mitochondria of the rat heart subjected to repeated 1 min and 10 min ischemic intervals. Values shown are initial concentrations (after a 20 min perfusion period, Group C) or at the end of protocols without 10 min ischemia (Group No I), with 10 min ischemia (Group I), and with 10 min ischemia plus infusion of 30 μM adenosine (Group I-Ado) during reperfusion. For the initial values, the control hearts (C) were perfused for 20 min. Values are mean ± SEM (*n* = 5–9). * *p* < 0.05 vs. concentration before 1 min ischemia. Respiration of Complex II was measured as the succinate-driven OCR. Complex I and Complex IV respiration were calculated as the resulting OCR after subtraction of rotenone or antimycin a driven respiration, respectively.

**Figure 7 pharmaceuticals-16-00599-f007:**
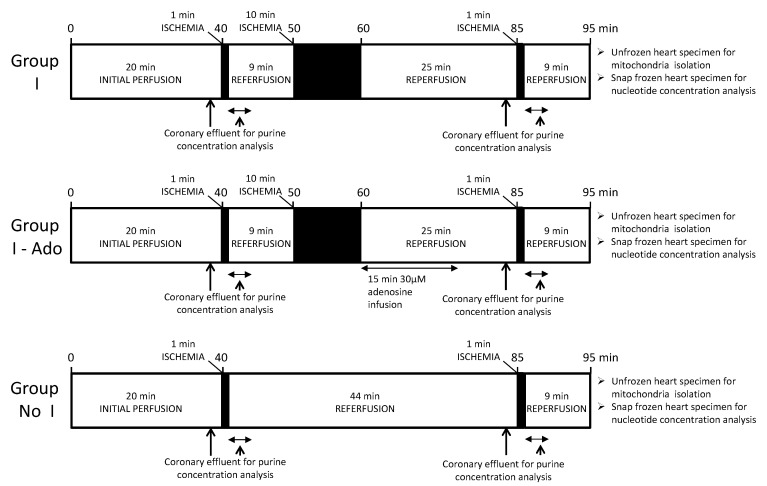
Experimental protocol.

## Data Availability

Data is contained within the article.

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
