# Peer review of "Hidden Pool of Cardiac Adenine Nucleotides That Controls Adenosine Production"

_pharmaceuticals, 2023, doi:10.3390/ph16040599_

Round 1

Reviewer 1 Report

This study demonstrates that the reduced production of adenosine and other nucleotide catabolites in the heart after repeated ischaemic events may be related to depletion of the adenine nucleotide pool rather than changes in the regulation of energy metabolism . 

The author showed that after transient administration of exogenous adenosine, endogenous purine catabolite production was considerably increased and this was associated with an increase in the cardiac adenine nucleotide pool. However, the large decrease in adenosine production with slight depletion of the cardiac nucleotide pool suggests that the entire pool is not available for nucleotide catabolism.

Adenosine, a purine isomer, is involved in vasodilation and is implicated in many cardioprotective mechanisms, and many reports on the regulation of its production have been published, and this paper is well received.

There are some minor amendments before publication.

1) The description of the figure is difficult to understand. The title of Figure 1 is small and its abbreviations are difficult to understand. Abbreviations such as I, I Ado, NoI and 40 min pre I, after I are very difficult to understand and an explanation would be easier for readers to understand. Figure 2 and 3 are easy to understand as explanations of abbreviations are indicated. 

2) In statistical analysis, it should be clearly stated how the data were analysed depending on whether they are parametric or non-parametric.

Reviewer 2 Report

The manuscript presented focused on an important clinical issue. However, there are several concerns regarding the methodological aspect and manuscript organization. In addition, in its current form, the study is descriptive only.

Title could be more informative and searchable.

Abstract is hard to follow and does not contain essential methodological information.

Manuscript needs grammar edits in many parts.

Although authors raised the concern about the lack of knowledge of “other factors contribute to the regulation of nucleotide catabolism and adenosine production”. Their study did not study the mechanisms of regulation of nucleotide catabolism and adenosine production.

No rationale was presented for why only one minute of ischemia was used and why they compared 40 and 85 min of perfusion before ischemia.

Authors overstated the study’s findings in many parts.

There is no standardization of figures in the manuscript. Different graph sizes are used in the same figure.

It is unclear why only Fig. 2 and 3 have a “C” group. It was not described what this group is.

Figure captions do not correspond to the information presented in the graphs.

To what do authors attribute the changes seen in the “No I” (control group)? It seems that there was not enough experimental control in these studies.

Although the results from 31P NMR spectra and contractile function are relevant to the study’s conclusion, the results are not shown.

There are several conflicts between the methods described and the description in the manuscript (mainly in the result section).

Additional experiments are necessary to support the authors’ conclusion.

Reviewer 3 Report

In this paper, Zabielska-Kaczorowska et al. present evidence Cardiac nucleotide pool and its role in post-ischemic adenosine production. The results of the study have been showing the maintenance of an intact cardiac adenine nucleotide pool is essential for the adequate adenosine response. The manuscript has some concerns. The reviewer makes the following suggestions:

1. This research requires other animal models to verify.

2. Provide siRNA and pharmacological analysis to examine the effects with target genes.

3. Provide mitochondrial physiological parameters function that should be included.

4. The discussion should be improved by correlating these results with the literature.

5. The manuscript is not linked to current conversations in the journal.

6. Most importantly, there is too little data to support the study.

Reviewer 4 Report

The article is dedicated to investigate the myocardial adenosine production in ischemic perfused rat hearts. The idea is interesting but fulfillment is definitely not the best. The article need very considerable revision and rewriting. At this stage it is difficult even to follow the results.

Major remarks:

1.      The Method part should be strongly corrected and written more understandable. Suggestion would be to make a scheme of how ischemia and reperfusion was done and when the samples were collected, what rat groups how were affected. Now everything is very confusing, for example, part 4.2 “Then, in I and I-Ado hearts were…’ What are those I and I-Ado? It should be very clearly stated what means all those groups and what exactly was done for them, i.e. how exactly ischemia and for how long was caused following by exact description of reperfusion. The number of rats in groups is not determined.  

Then another group of rats showed up – No-1 “In controls not subjected to 10 min ischemia (No-I), 1 min of ischemia was applied at 40 and 85 min.” For how long the ischemia and reperfusion was applied? 10 min or 1min?

What means “the balloon was deflated during 10 min ischemia”? Was it turned off? And then turned on? Why 40 and 85 min periods were chosen? How exactly the ischemia was caused, by turning off buffer and gas mixture? All procedures should be described and explained in details. Now it is difficult to follow results.

The English language should be strongly checked all over the manuscript. For example, section title “Tissue extraction for HPLC”. What tissue and from where was extracted?

The mitochondria isolation procedure is missing.

What samples were measured by NMR spectroscopy and HPLC? The methods of purine nucleotides, nucleosides, and bases such as adenosine measurements are missing.

2.      Result part. All abbreviations in legends should be clearly explained.

The description of results is very superficially written. Fig. 1 has no explanation in the text about what graph the authors are talking about. There is no detailed analysis of obtained data presented. Why there is a difference between 40 and 85 min? What is No-1 samples in Fig. 1? I - is ischemia of 1min? I-Ado is 1 min of ischemia and with adenosine? Probably, it should not be a surprise of better I-Ado group results since the external Ado was added.

3.      Fig. 2. What mean C if the authors state that “control hearts (No-I)”?

4.      The same remarks about the Fig 3. There are not explanations in the text about what graphs the authors are talking about.

5.      How the data in Fig 2 and 3 were normalized? As well as in other graphs?

6.      There are many confusing sentences such as in 2.6 part “It was increased, however, before ischemia at 85 min in groups I and I-Ado but not in group I.” Do the authors understand what they wrote?

7.      The new types of ischemia-reperfusion appear in fig 2,3,4,“ 1 min ischemia at 40 min of perfusion, 10 min ischemia at 50 min of perfusion and 1 min ischemia at 85 min of perfusion without (I) or with (I-Ado)” . There are too many variations that cannot be equally compared with each other if the ischemia and perfusion conditions vary. Moreover, these conditions are not shown in the graphs.

8.       The OCR analysis by Seahorse usually give much more parameters such as respiratory capacity, proton leak and many other. They are not shown. The broader and more detail analysis would give much more of additional information beside the total OCR. If the Seahorse measurements were not according to the manufacturers’ recommendations, it should be clearly described in the method part.

9.      „ 5. Conclusions“ part should be renamed into Discussion. The English language again should be very strongly checked, for example, it is written “ We had a considerable increase in endogenous purine catabolite production after the transient infusion of exogenous adenosine,…” I am confused, was it done with laboratory people or with rats…? Mitochondrial compartmentalization has another meaning than used in the discussion and is not experimentally proved as well as “mitochondrial pool”.  

The terms “produce purine catabolites” should be used with precaution, maybe it is just a release, not production. Discussion should be more constructive.

1  The concrete conclusions are missing.

Minor remarks:

1.      Chemical formulas such as CaCl2, CO2 and other should be written with subscripts.   

Round 2

Reviewer 2 Report

No further comments.

Author Response

Response: We are grateful for the new comments on our manuscript. Grammar editorial was made by Scribendi – Editing and Proofreading Services for English Documents.

Reviewer 3 Report

Photos of mitochondrial protein expression or cell staining must be provided to support the experimental results.

Author Response

We are grateful for the new comments on our manuscript. We agree that photos of mitochondrial protein expression or cell staining could support the experimental results of the publication. Unfortunately, we don't have any of these. We are able to perform additional experiments and attach photos to this or the next manuscript in about two months (to order such a large group of rats of the appropriate age from the animal facility). In our resources, however, we have rat heart tissue frozen using aluminium clamps at the end of the experiment, which can be used to perform ELISA tests of mitochondrial biogenesis, e.g. pgc1 alpha.

Reviewer 4 Report

The authors significantly improved the article and corrected the language and result description. However, more corrections are needed. The authors’ answers to the previous remarks are quite superficial.

Remarks:

1.      Fig. 1 and 2. have “Group C”, while in other figures it is missing.

2.      Fig. 2. The column design should be changed since the only white and black colors are present.

3.      The citation of “mitochondria isolation procedure” in the authors answers is No18, while in the article is already a citation No39. So, which one is correct?

4.      The used Seahorse kits should be clearly stated. It is not enough to write used Seahorse equipment. If the Mitostress kit was used, it has other types of data evaluation than it is shown in the article. So, the article is missing exact cat. No. of Seahorse kits and explanation how the activities of mitochondrial complex were evaluated. If not commercial kits were used, then the calculations should be explained.  

Author Response

    1. Fig. 1 and 2. have “Group C”, while in other figures it is missing. 

    Response: We are grateful for the new comments on our manuscript. For the initial values, control hearts (C) were perfused for 20 min. The authors used second control- no ischemic hearts (No-I) which were not subjected to 10 min global ischemia followed by reperfusion. Our main comparison was adenosine production at different time points. Figures 2, 3 and 5 were normalized for the initial values, and control hearts (C) were perfused for 20 min. Figures 1 and 4 were normalized for the No I group.

    1. Fig. 2. The column design should be changed since the only white and black colors are present.

    Response: Column design was changed.

    1. The citation of “mitochondria isolation procedure” in the authors answers is No18, while in the article is already a citation No39. So, which one is correct?

    Response: The manuscript was changed and now it is written: “Mitochondria isolation was performed according to the previous protocol [50]”.

    1. The used Seahorse kits should be clearly stated. It is not enough to write used Seahorse equipment. If the Mitostress kit was used, it has other types of data evaluation than it is shown in the article. So, the article is missing exact cat. No. of Seahorse kits and explanation how the activities of mitochondrial complex were evaluated. If not commercial kits were used, then the calculations should be explained.  

    Response: As Reviewer suggested, we corrected the Methods section (p.11, l.-331-351) by adding a detailed cat. No equipment and explaining necessary calculations. Moreover, we explained calculations in the Figure description(p.7, l.164-167).
